# Species Identification and In Vitro Antifungal Susceptibility of *Paecilomyces*/*Purpureocillium* Species Isolated from Clinical Respiratory Samples: A Multicenter Study

**DOI:** 10.3390/jof8070684

**Published:** 2022-06-29

**Authors:** Lorra Monpierre, Nawel Aït-Ammar, Isabel Valsecchi, Anne-Cécile Normand, Juliette Guitard, Arnaud Riat, Antoine Huguenin, Christine Bonnal, Boualem Sendid, Lilia Hasseine, Hélène Raberin, Marion Dehais, Stéphane Ranque, Christophe Hennequin, Renaud Piarroux, Eric Dannaoui, Françoise Botterel

**Affiliations:** 1Unité de Parasitologie-Mycologie, Département Prévention, Diagnostic, Traitement des Infections, CHU Henri Mondor, Assistance Publique des Hôpitaux de Paris (AP-HP), 94000 Créteil, France; lorrafelly.monpierre@aphp.fr (L.M.); nawel.ait-ammar@aphp.fr (N.A.-A.); 2DYNAMYC 7380, Faculté de Santé, Université Paris-Est Créteil (UPEC), 94010 Créteil, France; isabel.valsecchi@u-pec.fr (I.V.); eric.dannaoui@aphp.fr (E.D.); 3Laboratoire de Parasitologie-Mycologie, CHU La Pitié Salpetrière, AP-HP, 75013 Paris, France; annececile.normand@aphp.fr (A.-C.N.); renaud.piarroux@aphp.fr (R.P.); 4Sorbonne Université, Inserm, Centre de Recherche Saint-Antoine, CRSA, AP-HP, Hôpital Saint-Antoine, Service de Parasitologie-Mycologie, 75012 Paris, France; juliette.guitard@aphp.fr (J.G.); christophe.hennequin-sat@aphp.fr (C.H.); 5Division of Laboratory Medicine, Laboratory of Bacteriology, University Hospital of Geneva, 1205 Geneva, Switzerland; arnaud.riat@hcuge.ch; 6Laboratoire de Parasitologie-Mycologie, Pôle de Biopathologie, CHU de Reims, Université de Reims Champagne Ardenne, ESCAPE EA7510, 51100 Reims, France; ahuguenin@chu-reims.fr; 7Laboratoire de Parasitologie-Mycologie, CHU Bichat Claude Bernard, AP-HP, 75018 Paris, France; christine.bonnal@aphp.fr; 8Laboratoire de Parasitologie-Mycologie, CHU Lille, Inserm U1285-CNRS 8576, 59000 Lille, France; boualem.sendid@univ-lille.fr; 9Laboratoire de Parasitologie-Mycologie, CHU Nice, 06000 Nice, France; hasseine.l@chu-nice.fr; 10Laboratoire de Parasitologie-Mycologie, CHU Saint Etienne, 42270 Saint Etienne, France; helene.raberin@chu-st-etienne.fr; 11Laboratoire de Parasitologie-Mycologie, CHU de Rouen, 76000 Rouen, France; marion.dehais@univ-rouen.fr; 12Aix Marseille University, IRD, AP-HM, SSA, VITROME, IHU Méditerranée Infection, 13005 Marseille, France; stephane.ranque@ap-hm.fr; 13Unité de Parasitologie-Mycologie, Service de Microbiologie, Hôpital Européen Georges Pompidou, AP-HP, 75015 Paris, France; 14Faculté de Médecine, Université Paris-Sorbonne, 75006 Paris, France

**Keywords:** *Paecilomyces variotii*, *Paecilomyces maximus*, *Purpureocillium lilacinum*, antifungal susceptibility testing, molecular identification, MALDI-TOF mass spectrometry, MSI 2

## Abstract

*Paecilomyces* spp. are emerging fungal pathogens, where *Paecilomyces* *lilacinus* and *Paecilomyces variotii* are the most reported species. Taxonomic and phylogenetic revisions in this genus have shown that *P. variotii* represents a species complex, whereas *P. lilacinus* is related to another genus called *Purpureocillium*. The aims of this study were to identify clinical isolates of *Paecilomyces* spp. at the species level, and to determine their antifungal susceptibility profiles. 70 clinical *Paecilomyces* spp. isolates were identified by MALDI-TOF Mass Spectrometry (MS) and by multilocus rDNA genes sequencing including ITS and the D1/D2 genes. Among the 70 *Paecilomyces* spp. isolates, 28 were identified as *P. lilacinum*, 26 as *P. variotii stricto sensu*, and 16 as *P. maximus*. For antifungal susceptibility testing, Minimal Inhibitory Concentrations (MICs) or Minimal Effective Concentrations (MECs) were determined for 8 antifungals. All *P. lilacinum* isolates had high MICs and MECs of amphotericin B and echinocandins, respectively, unlike *P. variotii* and *P. maximus*. For azole drugs, MICs were molecule- and species- dependent. The differences in in vitro susceptibility to antifungals underline the importance of accurate species identification. The MALDI–TOF MS can be a good alternative in routine laboratory to ensure fast identification of *Paecilomyces* spp. and *P. lilacinum*.

## 1. Introduction

*Paecilomyces* spp. are hyaline moulds ubiquitously present in air, soil, decaying plants, and food products [1]. A small number of these fungi are opportunistic pathogens that can cause various diseases in humans and other mammals [2]. *Paecilomyces* fungi often infect immunocompromised patients, however immunocompetent patients are not spared and can become infected by direct inoculation of the fungus following trauma. Clinical presentations of *Paecilomyces* spp., such as cutaneous or catheter-related infections, ocular infections, peritonitis, sinusitis, pneumonia, osteomyelitis or fungemia have been reported in the medical literature [3,4,5,6,7].

Until the 2000s, *Paecilomyces variotii* and *Paecilomyces lilacinus* were the most commonly described species in *Paecilomyces* infections and colonisations affecting humans [3,6,7,8]. The two species show morphological similarities but can be differentiated based on conidial colour and growth rates. Over the last decade, molecular analyses showed that these two species are not related and ultimately belong to two different genera [9]. *P. variotii* belongs to the order Eurotiales, and *P. lilacinus* to the order Hypocreales, under a new family called *Ophiocordycipitaceae*. *P*. *variotti* currently represents a species complex including *P. variotii sensu stricto*, *Paecilomyces maximus*, *Paecilomyces divaricatus*, *Paecilomyces brunneolus*, and *Paecilomyces dactylethromorphus* [10]. Similarly, the nomenclature of *Paecilomyces lilacinus* was changed to *Purpureocillium lilacinum*, based on detailed phylogenetic analyses and partial gene sequencing of 18S rRNA [11].

Antifungal susceptibility data and treatment options for *Paecilomyces* and *Purpureocillium* infections are not well codified [2,12]. Available in vitro antifungal susceptibility data suggest significant differences in Minimal Inhibitory Concentration (MIC) ranges between species [13,14,15]. Therefore, accurate and early identification of *Paecilomyces* seems important to specify the culprit species, to predict its intrinsic resistance to antifungal agents, and to provide appropriate treatment, especially in high-risk patients.

Herein, we studied 70 respiratory isolates identified in a first instance as *Paecilomyces* spp. or *Purpureocilium lilacinum* based on their morphological characters and MALDI-TOF MS analysis. Species identification was further confirmed by sequence analysis of the intergenic transcribed spacer (ITS) regions, including the 5.8S rDNA, and the D1/D2 regions of 28S rDNA. Furthermore, the antifungal susceptibility profiles of *Paecilomyces*/*Purureocillium* species were also evaluated using EUCAST method.

## 2. Materials and Methods

### 2.1. Fungal Isolates

A total of 70 respiratory isolates of *Paecilomyces* spp. from patients were retrospectively collected between January 2002 and December 2018 in one Swiss and ten French university hospitals. Fungi were identified according to their morphological characters and/or MALDI-TOF MS in each center. All isolates were then sent to Creteil center and stored at −20 °C until use. The study was conducted in compliance with the ethical and legal requirements of the French law (15 April 2019) and the Declaration of Helsinki. Written or verbal informed consent from all participants was waived since isolates were collected as part of routine clinical work and patients’ identifiable information had already been anonymized prior to analysis.

### 2.2. Matrix-Assisted Laser Desorption Ionization-Time of Flight (MALDI-TOF) Mass Spectrometry Identification

All isolates were identified by MALDI-TOF MS using the Mass Spectrometry Identification 2 (MSI 2) platform database [16]. Fungal proteins were extracted from a mature subculture on Malt extract-agar medium (VWR, Rosny-Sous-Bois, France) using a previously described extraction protocol with minor modifications [17]. Briefly, a loop-full of mycelial colonies was transferred into a 1.5 mL microtube containing 300 μL of pure water and 900 μL of pure ethanol. After two centrifugations at 13,000× *g* for 2 min, the pellet was suspended in 25 μL of 70% formic acid and 25 μL of 100% acetonitrile. A sample of 1.0 μL of the fungal extract supernatant was spotted on a 96-spot polished steel plate in duplicate (Bruker Daltonics, Billerica, MA, USA) and allowed to completely dry at room temperature. Then, 1 μL of the IVD matrix HCCA portioned solution (Bruker Daltonik, Ref: 8290200, Billerica, MA, USA) was added. Protein spectra were analyzed using FlexControl^TM^ software with and the MBT compass software (Bruker, Billerica, MA, USA) and compared with Mass Spectrometry Identification (MSI) platform database 2. Identification was retained when the MSI score was above or equal to 20%. If several identification results were proposed for the same specimen, only the result with the best score was retained.

### 2.3. Molecular Identification and Phylogenetic Analysis

Molecular identification was performed by ITS DNA gene sequencing, including the 5.8S rDNA, and the D1/D2 gene regions, as recommended in previous studies for the identification of *Paecylomyces* spp. [18]. Whole genomic DNA was extracted from a mature subculture using a QIAamp DNA Mini Kit (Qiagen Sciences Ing., Courtaboeuf, France) after a step of beading in a MagNA Lyser instrument (Roche Diagnostics, Meylan, France). PCRs were performed in a 25 µL-final volume containing 1X HF buffer (ThermoFisher, Les Ulis, France), 100 µM of each deoxynucleosidetriphosphates (dNTPs), 1µM of each primer (i.e., ITS1 forward/ITS4 reverse [19] or D1/D2 NL-1 forward/NL-4 reverse [18]), 3% of DMSO, 1 unit of Phusion™High-Fidelity DNA Polymerase (ThermoFisher), and 50 ng of genomic DNA in a GeneAmp^®^ PCR system 9700 (Applied Biosystems™, Waltham, MA, USA). Sanger sequencing with BigDye ™ Terminator (ThermoFisher scientific) was performed at the Genomic platform of H. Mondor Biomedical Research Institute using the same primer pairs. The obtained sequences were analyzed using Chromas 2.6.6 software (Technelysium Pty Ltd., South Brisbane, Australia) and were compared with reference sequences retrieved from Westerdijk Fungal Biodiversity Institute database (https://wi.knaw.nl/ (accessed on 3 November 2021)) [20]. Genus and species level identifications were attributed using an identity score of ≥99% with respect to a reference entry. Multiple-sequence alignments were performed for each gene in MEGA 7.0.26 software (Auckland, New Zealand) using the CLUSTALW algorithm with manual adjustment. The phylogenetic tree was constructed using Maximum likelihood method and tested with 1000 rapid bootstrap inferences; it included sequences of the reference strain of each species and outgroup species sequences (Table 1).

### 2.4. Antifungal Susceptibility Testing

In vitro antifungal susceptibility testing was performed following the European Committee for Antimicrobial Susceptibility Testing (EUCAST) microdilution broth reference for filamentous fungi [21]. Eight antifungal agents were tested: amphotericin B (AMB), voriconazole (VRC), itraconazole (ITC), caspofungin (CAS) (Sigma-Aldrich, Saint-Quentin Fallavier, France), posaconazole (PCZ; MSD, Kenilworth, NJ, USA), isavuconazole (ISA; Basilea Pharmaceutica International Ltd., Basel, Switzerland), anidulafungin (AND; Pfizer Pharma New York, NY, USA), and micafungin (MCF; Astellas Pharma Inc., Tokyo, Japan). Each antifungal drug was tested in concentrations ranging from 0.016 to 8 mg/L. *Candida parapsilosis* ATCC 22019 and *Candida krusei* ATCC 6258 were included as quality controls. Results were read after 48 h of incubation at 37 °C. For polyenes and azoles AMB, VRC, ITC, PCZ, and ISA, the Minimal Inhibitory Concentrations (MICs) were determined both visually, as the lowest drug concentration leading to a complete inhibition of fungal growth (100% inhibition), and spectrophotometrically by optical density at 550 nm, using a 90% growth inhibition endpoint. On the other hand, for echinocandins CAS, MCF, and AND, the Minimal Effective Concentrations (MECs) were defined as the lowest concentration of the antifungal drug resulting in rounded and compact hyphal growth, compared with the unchanged fungal growth in the control without the antifungal drug.

For calculations, if the upper-limit drug concentration, i.e., 8 mg/L, did not inhibit fungal growth, a higher concentration of 16 mg/L was used.

## 3. Results

### 3.1. Isolates and Samples

Seventy clinical isolates of *Paecilomyces* spp. or *Purpureocillium lilacinum* taken from cultures of respiratory specimens of patients were retrospectively collected in many French regions and in a Swiss center. Of these isolates, 60% (*n* = 42) came from five Parisian (APHP) hospitals [Bichat (BCB), Henri Mondor (HMN), Pitie-Salpetriere (PSL), Saint Antoine (SAT), and European Georges Pompidou Hospitals (HEGP)]. The remaining 40% (*n* = 28) came from hospitals in the North (Lille, Rouen, Reims), and South-East (Saint-Etienne, Nice, Marseille (APHM)) of France, and from Switzerland (Geneva). The respiratory samples from which isolates were extracted included bronchoalveolar lavage fluids (30%, *n* = 21), bronchial aspirates (34%, *n* = 24), and sputa (36%, *n* = 25). None of these isolates were responsible for proven or probable invasive fungal infections. Therefore, they were considered as colonizers.

### 3.2. Species Identification

The MALDI-TOF MS identified 97% (*n* = 68/70) of the isolates at species level (identification scores ≥ 20). Species were *P. lilacinum* (*n* = 28/68, 41%), *P. variotii* (*n* = 25/68, 37%), and *P. maximus* (*n* = 15/68, 22%). The remaining two isolates could not be identified correctly at the species level by the MSI 2 database. Indeed, ITS and D1/D2 rDNA genes sequencing identified all isolates to the species level with the same identification. ITS and D1/D2 rDNA genes sequencing identified all 28 isolates of *P. lilacinum*, as did MALDI-TOF MS. For *P. variotii* species, one isolate was misidentified by MALDI-TOF MS and confused with *P. maximus.* For *P. maximus species*, one isolate was also not identified to species level by MALDI-TOF MS.

Phylogenetic relationships of the isolates and type strains species are illustrated in Figure 1 and Figure 2. Maximum likelihood trees, based on ITS and D1D2 sequences of isolates showed three distinct clades in *Paecilomyces maximus*.

### 3.3. Antifungal Susceptibility

The distributions of MICs/MECs of the different antifungals for each species are shown in Figure 3 and Figure 4. The results of the antifungal susceptibility testing (range, geometric mean MIC (GMIC), MIC_50_/MEC_50_, and MIC_90_/MEC_90_) on the 70 isolates are presented in Table 2. All *P. lilacinum* isolates showed growth inhibition at high MICs of amphotericin B (>8 mg/L) and high MECs of echinocandins (caspofungin, micafungin, and anidulafungin; >8 mg/L). In contrast, *P. variotii* and *P. maximus* were inhibited at low MICs of amphotericin B (≤0.5 mg/L) but their response to echinocandins was drug-dependant (Appendix A). For instance, MCF and AND were enough to inhibit both species due to their very low MECs (≤0.06 mg/L), whereas with caspofungin the MECs ranged from 0.01 to 16 mg/L. For azole drugs, the MICs were drug- and species- dependent. Overall, PCZ showed the best in vitro activity against *P. lilacinum* and *Paecilomyces* species, with MICs ≤ 0.5 mg/L. VRC and ISA were active in vitro against *P. lilacinum* (GMICs < 0.5 mg/L) but showed poor in vitro activity against *P. variotii* and *P. maximus* with GMICs of >4 mg/L. ITC showed intermediate antifungal activity against *P. lilacinum* with a GMIC of 1.414 mg/L and lower GMICs for *Paecilomyces* species.

## 4. Discussion

In recent years, we have observed the emergence of fungal infections in humans and animals caused by environmental moulds [2,12,22]. Among these moulds, *Paecilomyces* spp. and *Purpureocillium lilacinum* stand as ubiquitous, widely present in the environment, and are increasingly detected in respiratory samples of hospitalised patients. Such moulds are listed among the emerging agents that can cause localized fungal colonization, sometimes with other filamentous fungi, as well as infections in immunocompetent patients. Additionally, they can induce invasive infections especially in immunocompromised patients.

As for many filamentous fungi, the taxonomy of *Paecilomyces spp.* has evolved in recent years thanks to application of molecular methods. Currently, the genus *Paecilomyces* counts 10 species going from *Paecilomyces variotii sensu lato* complex (including *P. variotii stricto sensu*, *Paecilomyces brunneolus*, *P. maximus*, *Paecilomyces dactylethromorphus*, and *Paecilomyces divaricatus*), *Paecilomyces fulvus*, *Paecilomyces niveus*, *Paecilomyces tabacinus*, *Paecilomyces lagunculariae*, to *Paecilomyces zollerniae* [23]. Of the *P. variotii sensu lato* complex, *P. maximus*, previously called *P. formosus*, might represent in itself a complex of at least three genotypes [23]. However, to date, the three cryptic species have not been named yet. *Paecilomyces lilacinus* is now part of the new genus *Purpureocillium* which was first introduced by Luanga-Ard et al., in 2011 [11]. At present, the new genus includes five species named *P. lilacinum*, *Purpureocillium lavendulum, Purpureocillium takamizusanense, Purpureocillium roseum*, and *Purpureocillium atypicola* [24,25,26]. The latest taxonomic changes considerably complicate the identification of *Paecilomyces* spp. or *Purpureocillium* spp. in routine work of clinical microbiology laboratory. To our knowledge, few studies conducted in the field of clinical mycology have focused on the diversity of *Paecilomyces* species isolated from human samples, especially in the context of respiratory colonization [13,15,18]. The extremotolerant nature is thought to contribute to the pathogenic potential of these fungi whose presence in clinical samples might be explained by their omnipresence in food and indoor environments. Additionally, being immunodepressed may prompt respiratory colonization to become infection as it is the case with other moulds. In this context, we studied the distribution of *Paecilomyces* species and determined their in vitro susceptibility to the most common antifungal agents. In our study, *P. variotii sensu lato* was the most common species (*n* = 42; 60%) followed by *P. lilacinum* (*n* = 28; 40%).

Of the *P. variotii sensu lato* complex, only *P. variotii stricto sensu* and *P. maximus* were identified by both ITS and D1/D2 rDNA genes sequencing. The choice of molecular targets for the identification of *Paecilomyces* spp. and *Purpureocillium* spp. was based on the recommendations proposed by the ECMM in cooperation with ISHAM and ASM in 2021 [2].

Our results suggest using MALDI-TOF MS as a good tool to identify *Paecilomyces* spp. and *Purpureocillium lilacinum* in routine laboratory work. The concordance between molecular and MALDI-TOF MS scores reached 100% for *P. lilacinum*, whereas it was 96% for *P. variotii stricto sensu* and 94% for *P. maximus* on MSI 2 database platform as compared with the reference sequencing method (Appendix A). However, the performance of MALDI-TOF MS for mould identification is database-dependent [18,27]. In 2014, Barker et al. [18] evaluated the performance of MALDI-TOF MS in the identification of 77 genetically confirmed isolates of *Paecilomyces* species. He showed the interest of using MALDI-TOF reference library for identifications. In their studies, the agreement between the molecular and proteomic methods was 92.2% only after supplementation of MALDI-TOF MS database with type strains.

In our study, MALDI-TOF MS misidentified two isolates, one *P. variotii* and one *P. maximus.* With the former, the MSI 2 database was not able to distinguish whether it was *P. variotii* or *P. maximus* because the two scores were related, whereas the latter was identified as *P. maximus* with an MSI 2 score lower than 20%. These results suggest the need for continuous updating of MALDI-TOF MS databases to obtain better performance, i.e., accurate identification of *Paecilomyces* spp.

Significant differences in in vitro antifungal susceptibility were observed between *P. variotii sensu lato* and *P. lilacinum*, which is consistent with the findings of other studies [4,5,6,8,13,14,15]. In general, the antifungal susceptibility profiles of *P. variotii stricto sensu*, *P. maximus*, *P. dactylethromorphus*, and *P. divaricatus* appeared to be similar to what other studies had already shown, although a limited number of isolates were tested in those studies [13,14,15]. Previous data demonstrated that *P. lilacinum* had its growth inhibited at higher in vitro MICs/MECs of amphotericin B, itraconazole, and echinocandins as compared with *P. variotii sensu lato*, which is generally more susceptible. On the contrary, high MICs of voriconazole were needed for *P. variotii* but not for *P. lilacinum* [4,5]. In our study, all azoles were active on *P. lilacinum*, even ITC which showed a wider range of MICs. On the other hand, all *P. lilacinum* isolates had elevated high MICs of AMB (>8 mg/L), suggesting an intrinsic resistance. These results are particularly interesting since few filamentous fungi are naturally resistant to AMB, a broad-spectrum antifungal agent with fungicidal activity. According to the literature, only a few fungal species are resistant to AMB, such as *Aspergillus terreus*, *Aspergillus tanneri*, *Fusarium* spp. or *Lomentospora prolificans*, whose molecular resistance mechanism is mostly unknown [28]. However, it is now known that the resistance mechanism of *A. terreus* to AMB is complex and multifaceted [29,30,31].

*P. variotii* and *P. maximus* showed a different susceptibility profile with a GM of MICs/MECs of <0.5 mg/L of AMB, ITC, PCZ, MCF, and AND. The MECs of CAS were variable as we used GMECs of 4.095 mg/L for *P. variotii* and GMECs of 0.740 mg/L for *P. maximus.* In contrast, VRC and ISA were not active against *P. variotii* and *P. maximus* (GMICs > 4 mg/L). According to our results, *Paecilomyces* spp. showed resistance to both VRC and ISA. Recently, some studies have shown a resistance to voriconazole in moulds such as *Rasamsonia* complex or *Acremonium* spp. [32], and in both, the resistance mechanism is still unknown. These data suggest that azole resistance concern not only *Aspergillus* spp. but also other environmental moulds. To our knowledge, our study is the first to have investigated the in vitro sensitivity of isavuconazole in *P. variotii strains* [4]. The mechanism of cross-resistance to voriconazole and isavuconazole in *P. variotii* and *P. maximus* remains to be explored.

In conclusion, in the respiratory samples we used in our study, only *P. lilacinum*, *P. variotii stricto sensu*, and *P. maximus* were found. The variability in the observed in vitro susceptibilities to antifungal drugs underlines the importance of precise and correct fungal identification at the species level in order to optimize treatment of *Paecilomyces* spp. or *Purpureocillium lilacinum*–related infections. The therapeutic difficulties originate from the in vitro resistance of *P. lilacinum* to AMB and echinocandins, and from the resistance of *P. variotii* and *P. maximus* to azoles. Eventually, MALDI–TOF MS proved to be a rapid and reliable alternative to use in routine identification of *Paecilomyces* spp. and *P. lilacinum* in clinical mycology laboratories.

## Figures and Tables

**Figure 1 jof-08-00684-f001:**
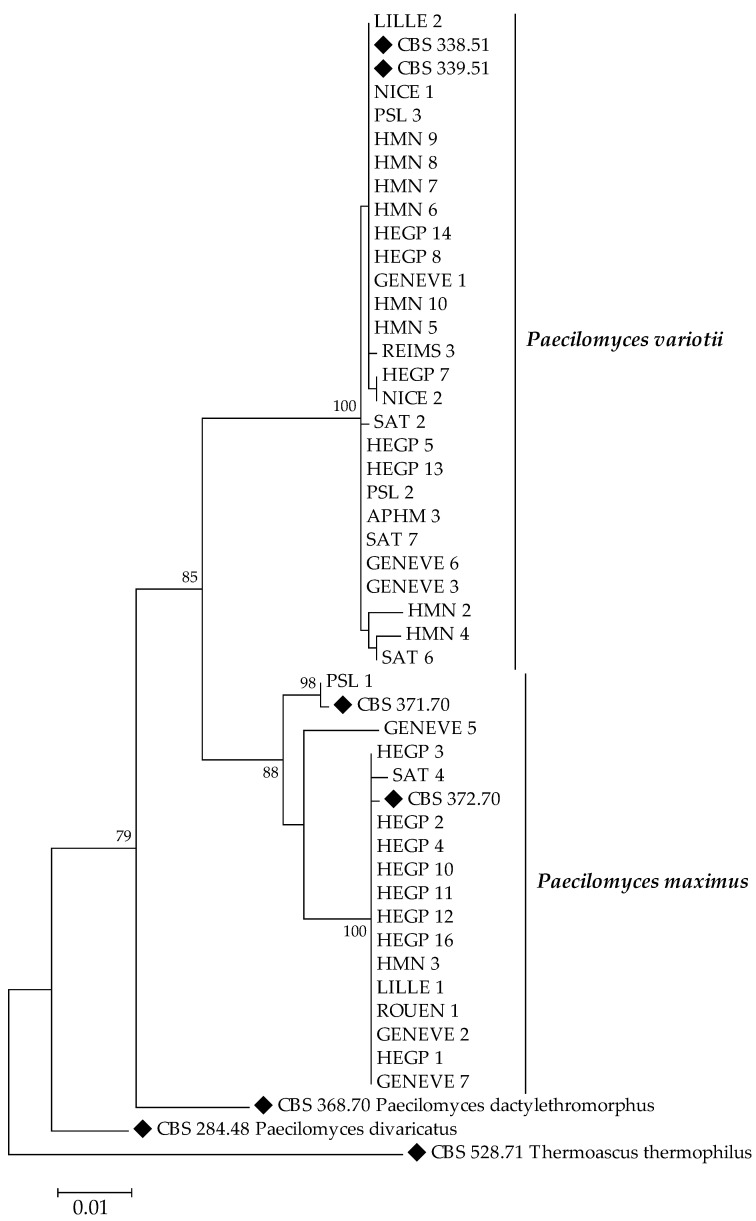
Maximum Likelihood tree of *Paecilomyces* spp. based on combined data set of ITS and D1/D2. Numbers above the nodes represent bootstrapping values generated from 1000 replicates, using a Kimura 2-parameter model. Only values above 70% are indicated. *Thermoascus thermophilus* CBS 528.71 has been used as the outgroup.

**Figure 2 jof-08-00684-f002:**
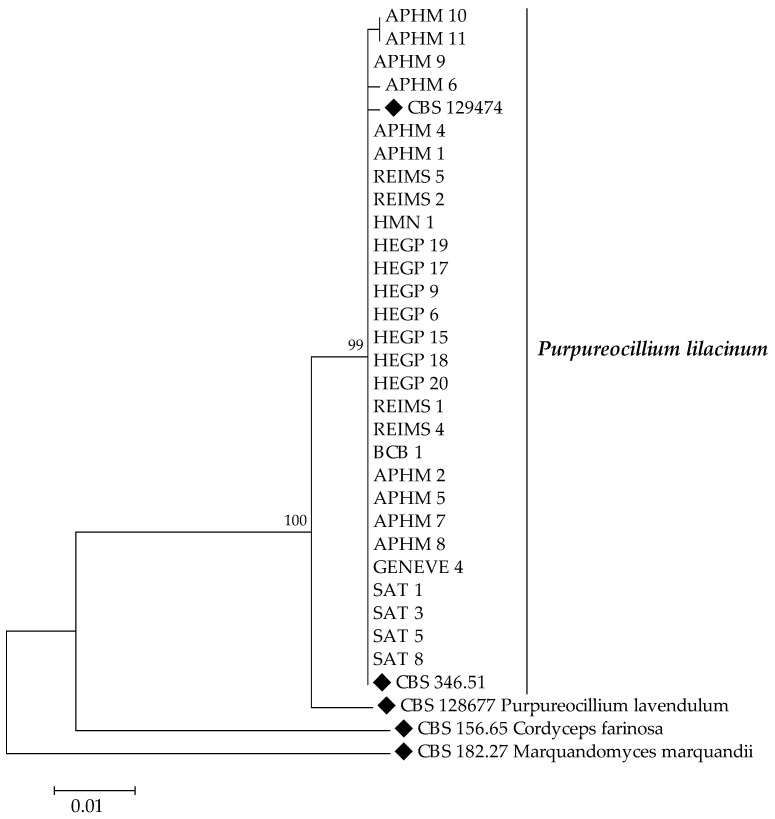
Maximum Likelihood tree of *Purpureocillium* spp. based on combined data set of ITS and D1/D2. Numbers above the nodes represent bootstrapping values generated from 1000 replicates, using a Kimura 2-parameter model. Only values above 70% are indicated. *Cordyceps farinosa* CBS 156.65 and *Marquandomyces marquandii* CBS 182.27 have been used as outgroups.

**Figure 3 jof-08-00684-f003:**
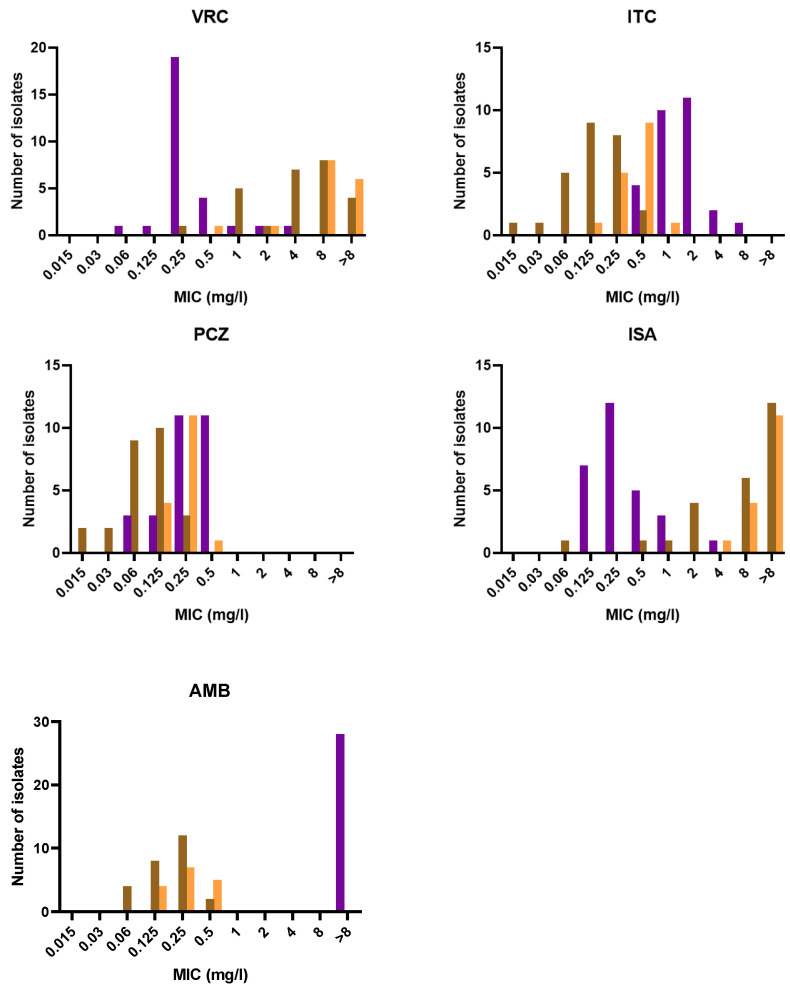
Distributions of azoles and amphotericin B MICs for *P. lilacinum* (purple), *P. variotii* (brown) and *P. maximus* (orange). **VRC**: voriconazole; **PCZ**: posaconazole; **ITC**: Itraconazole; **ISA**: Isavuconazole, **AMB**: amphotericin B.

**Figure 4 jof-08-00684-f004:**
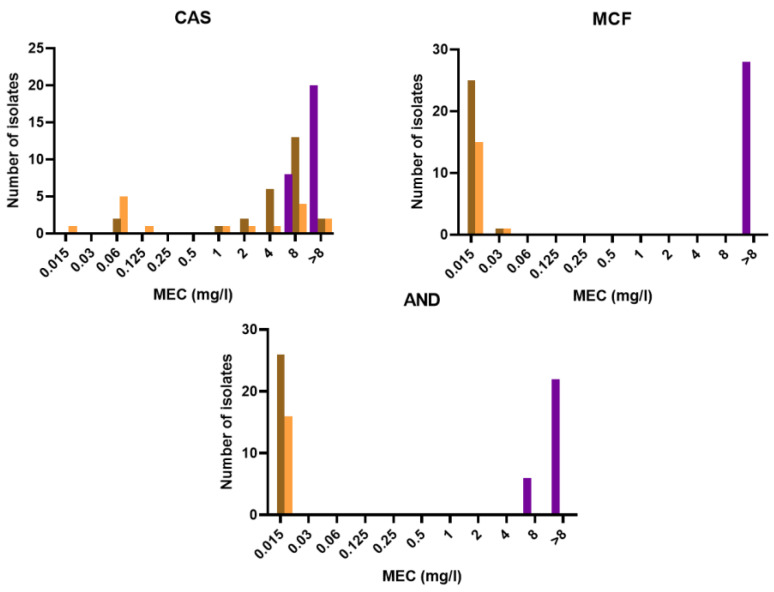
Distributions of echinocandins MECs for *P. lilacinum* (purple), *P. variotii* (brown) and *P. maximus* (orange) **CAS**: caspofungin. **MCF**: micafungin; **AND**: anidulafungin.

**Table 1 jof-08-00684-t001:** Sequences reference used in this study [20].

CBS No		Gene Bank Accession N°	
	Species	ITS [20]	D1D2 [20]
CBS 372.70 ^T^	*Paecilomyces maximus*	MH859719.1	MH871470.1
CBS 371.70 ^T^	*Paecilomyces maximus*	MH859718.1	MH871469.1
CBS 339.51	*Paecilomyces variotii*	MH856887.1	MH868409.1
CBS 338.51	*Paecilomyces variotii*	MH856886.1	MH868408.1
CBS 284.48 ^T^	*Paecilomyces divaricatus*	MH856344.1	MH867896.1
CBS 368.70	*Paecilomyces dactylethromorphus*	MH859715.1	MH871467.1
CBS 129474	*Purpureocillium lilacinum*	MH865347.1	MH876802.1
CBS 346.51	*Purpureocillium lilacinum*	MH856891.1	MH868413.1
CBS 528.71 ^T^	*Thermoascus thermophilus*	MH860254.1	MH872018.1
CBS 128677 ^T^	*Purpureocillium lavendulum*	MH864976.1	MH876429.1
CBS 182.27 ^T^	*Marquandomyces marquandii*	MH854923.1	MH866418.1
CBS 156.65	*Cordyceps farinosa*	MH858528.1	MH870163.1

^T^: Type species.

**Table 2 jof-08-00684-t002:** Results of in vitro antifungal susceptibility testing on the 70 isolates using the EUCAST method.

Species and Drug	Range	GM	MIC_50_/MEC_50_	MIC_90_/MEC_90_
	(mg/L)	(mg/L)	(mg/L)	(mg/L)
*Purpureocillium lilacinum* (*n* = 28)				
VRC	0.06–4	0.320	0.25	0.5
ITC	0.5–8	1.414	1	2
PCZ	0.06–0.5	0.262	0.25	0.5
ISA	0.125–4	0.305	0.25	1
CAS	8–16	13.125	16	16
MCF	16–16	16	16	16
AND	8–16	13.792	16	16
AMB	16–16	16	16	16
*Paecilomyces variotii stricto sensu* (*n* = 26)				
VRC	0.25–16	4.108	4	16
ITC	0.015–0.5	0.130	0.125	0.25
PCZ	0.015–0.25	0.080	0.06	0.125
ISA	0.06–16	5.499	8	16
CAS	0.06–16	4.095	8	8
MCF	0.015–0.03	0.015	0.015	0.015
AND	0.015–0.015	0.015	0.015	0.015
AMB	0.06–0.5	0.171	0.25	0.25
*Paecilomyces maximus* (*n* = 16)				
VRC	0.5–16	8.775	8	16
ITC	0.125–1	0.379	0.5	0.5
PCZ	0.125–0.5	0.228	0.25	0.25
ISA	4–16	12.699	16	16
CAS	0.015–16	0.746	1	8
MCF	0.015–0.06	0.016	0.015	0.015
AND	0.015–0.015	0.015	0.015	0.015
AMB	0.125–0.5	0.262	0.25	0.5

GM: Geometric Mean; MIC: Minimum Inhibitory Concentration; MEC: Minimum Effective Concentration; VRC: Voriconazole; ITC: Itraconazole; PCZ: Posaconazole; ISA: Isavuconazole; CAS: Caspofungin, MCF: Micafungin, AND: Anidulafungin; AMB: Amphotericin B.

## Data Availability

The obtained sequences were submitted to GenBank under accession numbers ON853835 to ON853904 and ON853920 to ON853989 for ITS and D1/D2, respectively.

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
