# Peer review of "Species Identification and In Vitro Antifungal Susceptibility of Paecilomyces/Purpureocillium Species Isolated from Clinical Respiratory Samples: A Multicenter Study"

_jof, 2022, doi:10.3390/jof8070684_

Round 1
Reviewer 1 Report
Comments
1) Line 123 and line 169
Neighbor-joining is not a method that allows phylogenetic inference, it only estimates evolutionary distances, so I recommend to write "Neighbor-joining tree" instead "pylogenetic tree", which is conceptually correct.
2) The Neighbor-joining tree could or could be not included in the manuscript, depending authors choice. If the authors prefer to present the tree, they should include at least an additional species of the genus Purpureocillium, I suggest Purpureocillium lavendulum.
Reviewer 2 Report
The work of Monpierre L. and cols. is a work aimed to identify clinical isolates of Paecilomyces/Purpureocillium spp. at the species level, and to determine their susceptibility profiles to 8 antifungals. In general, the experimental design of the work, as well as the adopted methods were adequate and consistent with the planted objectives. Moreover, the manuscript is well written and adequately structured.
Major revisions are required:
- Nothing is known about clinical relevance of the isolates – It is important to specify the particular respiratory origins of the analyzed isolates (lung biopsy, BAL, tracheal aspirate, sputum?), as well as if the fungal isolations were contamination, colonization or proven agent of infection.
- The point 2.3 “Molecular identification and phylogenetic analysis” must include the thermocycler utilized for the molecular amplifications.
- The obtained sequences were compared with reference sequences retrieved from NCBI nucleotide database (BLAST), but many of that deposits haven’t a phenotypic support by Mycologists, so I strongly recommend to use ISHAM ITS (http://its.mycologylab.org/) and MycoBank (http://mycobank.org/) online databases.
- Phylogeny: Neighbor-joining (NJ) analysis rather obsolete, I would recommend to use Maximum likelihood, Bayesian inference or Maximum parsimony analysis.
- Phylogeny: The constructed tree must be rooted including at least one outgroup species.
- The phylogenetic tree has spliced labels.
- Table 2 could be omitted.
- More discussion regarding the results of antifungal susceptibility is needed.
Reviewer 3 Report
Thank you for inviting me to evaluate the article titled “Species Identification and in vitro Antifungal Susceptibility of Paecilomyces/Purpureocillium Species Isolated from Clinical Respiratory Samples: a Multicenter Study”. The aims of this study were to identify seventy clinical isolated Paecilomyces spp. from different hospitals and to determine their antifungal susceptibility profile. I think it has certain guiding significance for the selection of clinical medication. However, if only the origin of the Paecilomyces spp. were different, this study should not be called " a multicenter study". Except the definition confusion, the figures and tables in this manuscript should be further improved. The following are the questions in this manuscript:
- The author should remove the description of "a multicenter study" from the title of the manuscript. Clearly, the description in the methodology section shows that this study was not multicenter research.
- I recommend that Figure 2 and Figure 3 can be combined into one Figure, since they are describing the same kind of experimental results.
- Some information overlapped between Table 3 and Figures 2-4. Can Table 3 be submitted as a supplementary document? In addition, the authors should further explain why only "CAS, MCF and AND" are selected for MEC experiment in this manuscript.
- In my opinion, MEC experiment should present representative images to support Figure 4.
Round 2
Reviewer 2 Report
The authors addressed all the previous suggestions in the revised version of the manuscript. I recommend acceptance in its present form.